An effective transformer based on dual attention fusion for underwater image enhancement

Hu Xianjie
Liu Jing 20060034@kust.edu.cn
Li Heng
Liu Hui
Xue Xiaojun
Faculty of Information Engineering and Automation, Kunming University of Science and Technology , Kunming , China
Somani Arun
Electronic publication date: 2024 Apr 30
Publication date: 2024
Volume: 10
Electronic Location ID: e1783
Received 2023 Aug 8; Accepted 2023 Dec 6
Copyright: © 2024 Hu et al.
Copyright year: 2024
Copyright holder: Hu et al.
License: This is an open access article distributed under the terms of the Creative Commons Attribution License, which permits unrestricted use, distribution, reproduction and adaptation in any medium and for any purpose provided that it is properly attributed. For attribution, the original author(s), title, publication source (PeerJ Computer Science) and either DOI or URL of the article must be cited.
License URL: https://creativecommons.org/licenses/by/4.0/

Keywords: Artificial intelligence, Computer vision, Neural networks, Underwater image enhancement

Funding: National Natural Science Foundation of China 62263016 Applied Basic Research Foundation of Yunnan Province 202001AT070038 This work was supported by the National Natural Science Foundation of China (No. 62263016) and the Applied Basic Research Foundation of Yunnan Province (No. 202001AT070038). The funders had no role in study design, data collection and analysis, decision to publish, or preparation of the manuscript.

==============================
Underwater images suffer from color shift, low contrast, and blurred details as a result of the absorption and scattering of light in the water. Degraded quality images can significantly interfere with underwater vision tasks. The existing data-driven based underwater image enhancement methods fail to sufficiently consider the impact related to the inconsistent attenuation of spatial areas and the degradation of color channel information. In addition, the dataset used for model training is small in scale and monotonous in the scene. Therefore, our approach solves the problem from two aspects of the network architecture design and the training dataset. We proposed a fusion attention block that integrate the non-local modeling ability of the Swin Transformer block into the local modeling ability of the residual convolution layer. Importantly, it can adaptively fuse non-local and local features carrying channel attention. Moreover, we synthesize underwater images with multiple water body types and different degradations using the underwater imaging model and adjusting the degradation parameters. There are also perceptual loss functions introduced to improve image vision. Experiments on synthetic and real-world underwater images have shown that our method is superior. Thus, our network is suitable for practical applications.

Introduction

Underwater images can provide intuitive and detailed information about marine life and marine ecology, and have significant contribution to development and utilization of marine resources. However, the imaging process is disturbed by a variety of physical properties in the water, which leads to considerable degradation of the images. Specifically, the light wavelength-dependent attenuation due to the water absorbing selectively to light causes the color shift of images (Akkaynak et al., 2017). Moreover, the backscattering of light by particles in the water causes hazy scenes and low image contrast (Schettini & Corchs, 2010). This image degradation increases the difficulty of extracting information from it. The development of underwater computer vision and robotic visual perception (Islam, Xia & Sattar, 2020) is limited by the incapacity of degraded images to provide rich and accurate information for advanced vision tasks. Therefore, underwater image enhancement techniques are of significant importance in exploring the ocean world (Yuh & West, 2001).

Techniques for enhancing underwater images generally separate into three categories: physical model-based, non-physical model-based, and data-driven methods. Specifically, the physical model-based method reverses the underwater imaging model to obtain clear images. This method uses valid priori assumptions (Drews et al., 2016) to estimate imaging parameters, including medium transmission and background light. However, the robustness of this method is affected because the physical imaging models cannot adapt to complex and diverse underwater environments (Liu et al., 2020). What’s more, the accurate estimation of multiple imaging parameters is a challenge since a highly complex underwater imaging model. Additionally, underwater imaging models are unnecessary for non-physical model-based methods, which improve image quality by directly altering pixel values, such as adjusting image contrast, brightness, and saturation. However, since the underwater image formation model is ignored, it introduces artificial effects such as noise and color bias.

Over the past decade, deep learning techniques have developed rapidly. The remarkable progress of deep learning techniques in low-level vision problems (Dong et al., 2015; Zhu et al., 2017) leads to their gradual application in underwater image enhancement (Li et al., 2017; Wang et al., 2017). The data-driven method extracts image features using a network structure to discover data relevance, which learns to map raw images to their clear images. However, existing underwater image datasets are small in scale and have few underwater scenes, result in the performance of the data-driven approach is limited.

Convolution can aggregate local information well, but it is hard to model global long-range dependencies. In reality, underwater images exhibit inconsistent attenuation in different color channels and spatial regions. Since the convolution uses the same operation in space, it is not the optimal solution for this spatially different attenuation.

The farther the object is from the camera, the more severe the degradation of the underwater image, as described by the underwater image formation model. The image exhibits spatially inconsistent attenuation due to the different areas of the image having different distances from the scene to the camera. Therefore, the network should have global aggregation capability to obtain global information about the scene and thus estimate the correct scene depth. In addition, the selective absorption of light by water resulting in image color bias causes significant differences in color channel values. Since the weights of these channel information are different, the network should pay attention to the channel information that causes color degradation to better correct the image color. Moreover, the local contextual information is required to reconstruct clear images, and relying solely on Transformer lacking inductive biases may limit performance. However, Convolution and Transformer have their drawbacks. Convolution is not good at modeling remote dependencies while Transformer is not good at aggregating local information. The combination of local attention and non-local attention is an appropriate approach. It is common to encounter spatially inconsistent color attenuation in larger-scale images, whereas smaller-scale images generally exhibit relatively more balanced color attenuation in space. When underwater images exhibit spatially inconsistent color attenuation, non-local operations, such as remote dependency modeling, can be more helpful for color correction. However, when the color attenuation in an image is approximately consistent in space, local operations, such as convolutions with shared parameter characteristics, can not only correct colors but, more importantly, are more helpful for image detail reconstruction. We aim to investigate how to balance local attention and global attention in underwater image enhancement to better enhance the color and details of the image.

In light of the aforementioned problems, this article proposes a network based on fusion attention block (FAB) for underwater image enhancement. The main purpose of network design is to globally correct the distorted color of the image and reconstruct the texture of image details. U-Net can extract multi-scale features of the image, including both global and local information, which makes it suitable for our network. The global information of the image contributes to the correction of color, while the local information contributes to the restoration of image details. However, using U-Net alone to capture global information has limited capability, while Swin Transformer block (STB) (Liu et al., 2021) inherently excels at capturing global information. Therefore, we employ STB for global dependency modeling. However, Transformers lack the locality of convolution, which results in their not good at local modeling. Therefore, we introduce residual convolutions to perform local attention. We perform global attention and local attention in parallel on two branches, and adaptively fuse their results. In addition, certain wavelengths of light, such as red, experience faster attenuation in water. Therefore, we also introduce a channel attention mechanism to increase attention to channels that suffer from severe attenuation.

To be concrete, our model is based on the U-Net (Ronneberger, Fischer & Brox, 2015) that includes FAB inserted as the base building blocks. Therefore, it allows the model adaptively select non-local and local attention at different scales to benefit image feature extraction and reconstruction. Our contributions are summarized as follows:

1) We propose a novel network model for underwater image enhancement by incorporating fusion attention blocks into U-Net to boost the modeling capability of the network. It can adaptively fuse non-local and local features to better correct image color and reconstruct image details.

2) We propose a new synthetic underwater image dataset utilizing the underwater image formation model. The synthetic dataset contains simulated underwater images of various degradation types and can be used to train underwater image enhancement model.

3) We conduct a series of experiments to prove that the proposed method has the superior performance. Thus, our method can be applied in practice.

Background knowledge and related work

Underwater image formation model

There are similarities between foggy weather images and blurry underwater images. As a result, the imaging in underwater environments is traditionally described by the atmospheric scattering model (Frenkel, 1924). However, it is only applicable in some scenes which shallow water scenes with low backscattering as the atmospheric scattering model is not suitable for all the diverse and complex underwater scenes in the real world. This results in a significant deviation from the actual underwater imaging model. Therefore, most existing approaches usually use an improved model (Chiang & Chen, 2011) to describe the underwater imaging process. Compared with the atmospheric scattering model, the selective attenuation of different wavelengths of light in water is considered by it, thus improving its generalization. The improved model of underwater imaging can be expressed as

(1) Uλ(x)=Iλ(x)⋅Tλ(x)+Bλ⋅(1−Tλ(x))

where x represents a specific point in the underwater scene and λ denotes the wavelength of the light at that point, including red, green and blue colors. The underwater image captured by the camera is denoted as Uλ(x), while Iλ(x) represents the clear scene image. Bλ denotes the global background light component. Tλ(x) denotes the remaining energy ratio of the light radiation from the scene points reaching the camera, resulting in light scattering and color shift, which is expressed as

(2) Tλ(x)=Eλresidual(x)Eλinitial(x)=Nrer(λ)d(x)

where Eλresidual and Eλinitial represent the residual energy and initial energy, respectively, of the light with wavelength λ that at the camera and the scene point. while d(x) is the distance traveled by light from scene point x. Nrer(λ) is the normalized residual energy ratio. It describes the ratio of the residual energy of light propagating per unit distance in the medium, which is related to the wavelength of the light. For water media, the red light with longer wavelengths attenuates more rapidly than the shorter wavelengths of blue and green, resulting in underwater images that usually appear blue or green in style.

Underwater image enhancement

Physical model-based methods first assume an imaging model. Then, physical parameters within the imaging model are estimated using prior methods. Finally, the imaging model is reversed to get a clear image. The core problem is to estimate the two parameters of transmittance and global background light by using priors. Existing priors include underwater dark channel priors (Drews et al., 2013), red channel prior (Galdran et al., 2015), minimum information prior (Li et al., 2016), underwater light attenuation prior (Song et al., 2018), etc. The underwater dark channel prior method (Drews et al., 2013) improves the DCP method (He, Sun & Tang, 2010) to better adapt to the underwater scene, but it only considers the blue and green channels, while the information in the red channel is ignored. The method of image blurriness and light absorption, suggested by Peng & Cosman (2017), can estimate depth maps for underwater images. The underwater light attenuation prior (Song et al., 2018) can effectively estimate scene depth, which assumes that it is closely related to the difference between the red channel and the maximum value of the channels blue and green. These physical model-based methods can produce good results when the imaging model and priors correspond with the underwater scene. But it does not generalize well as the difficulty of adapting to varied underwater scenes. Moreover, the resulting image has poor visual effects because it ignores the vision perception characteristics of the human eye.

Non-physical model-based methods enhance the image primarily through adjustments to its brightness and contrast. They include conventional methods used earlier, such as histogram equalization (Ghani & Isa, 2015; Hitam et al., 2013) to improve contrast and white balance (Iqbal et al., 2010, 2007) to correct color. In addition, there are fusion based methods (Ancuti et al., 2012) and Retinex based methods (Fu et al., 2014; Zhang et al., 2017). Ancuti et al. (2012) proposes an image fusion method to enhance underwater images by fusing two images with corrected color and improved contrast using different weight maps. Zhang et al. (2017) proposed a multi-scale Retinex method utilizing the characteristics of the three channels of the CIELab color space. The non-physical model-based method has a significant effect on improving the brightness and contrast of underwater images. However, since underwater images have spatially inconsistent attenuation, it is not reasonable to globally correct the color of the images. Because the physical imaging process is not considered, the effectiveness of its enhancement is limited. Moreover, it often introduces artificial enhancement effects.

Since outstanding results have been seen for computer vision tasks through the application of convolutional neural networks (CNNs), several works have employed CNNs to enhance underwater images (Anwar, Li & Porikli, 2018; Fabbri, Islam & Sattar, 2018; Islam, Xia & Sattar, 2020; Li et al., 2019a). Anwar, Li & Porikli (2018) proposed the UWCNN network, which consists of densely connected convolutional layers. In addition, they based on the underwater imaging model and different water-type parameters to synthesize an underwater image dataset. Li et al. (2019a) proposed a Water-Net to enhance underwater images, which incorporates a gated fusion architecture. Three images are used as inputs to the network, which are obtained by applying gamma correction, histogram equalization and white balance to the underwater images respectively. It combines the input images according to the learned confidence map to construct the enhanced image. In addition, some methods utilize the concept of game theory to propose a generative adversarial network. Islam, Xia & Sattar (2020) propose a GAN approach to enhance underwater images. Its generator is based on U-Net, and the discriminator uses the PatchGAN (Isola et al., 2017) structure, which has fewer computational parameters and can better capture high-frequency features. Fabbri, Islam & Sattar (2018) proposed UGAN, which uses synthetic data for network training. The training images are generated by CycleGAN (Zhu et al., 2017) converting underwater images from clear to blurry.

Although the above methods achieve better results compared to physical and non-physical model-based methods, they also have some shortcomings. The enhancement results generated by the GAN-based model are unstable, although it has excellent visual perception. The convolution has parameter sharing, which makes it inflexible to adapt to input features. Furthermore, it cannot model long-range dependencies because of limited receptive fields. The Transformer (Vaswani et al., 2017) is a Seq2Seq model devised for processing sequences within the domain of natural language processing (NLP), which was later utilized for computer vision. Its self-attention mechanism can effectively capture global dependencies between data. Moreover, unlike convolution, the self-attentive mechanism allows the image content to interact with the attention weights. With these advantages, it is extensive applications in various visual tasks including image denoising (Liang et al., 2021), image segmentation (Strudel et al., 2021), image recognition (Dosovitskiy et al., 2020) and object detection (Carion et al., 2020). Dosovitskiy et al. (2020) reshape an image into a sequence of image patches and feed it into a pure Transformer model. Its excellent performance on image classification tasks shows that the dependence on computer vision on CNN is unnecessary. Liu et al. (2021) proposed a Swin Transformer network, which uses a shifted window strategy to limit the attention calculation range. It not only has higher computational efficiency but also has the locality of the convolution operation. Peng, Zhu & Bian (2023) proposed a U-shape network that introduces Transformer into a GAN for underwater image enhancement. The color channels with severely attenuated are strengthened by the inter-channel self-attention of the network. It also computes spatial global attention between image patches to focus on the regions with severe degradation. Compared to previous underwater image enhancement methods, our network focuses not only on more severely attenuated spatial areas and color channels but also on local modeling that is beneficial for image recovery to reconstruct refined details.

Proposed method

Our goal is to develop an efficient Transformer model for underwater image enhancement with good robustness. First, the suggested overall architecture is presented. After that, the important components of the network are introduced, including the channel attention block, local attention block and adaptive feature fusion block. Finally, the loss function for training is introduced.

Network architecture

The overall architecture of the proposed network is presented in Fig. 1. Components of the overall pipeline including encoders, bottlenecks, and decoders. The decoder receives information through skip-connection from the encoder.

Figure 1 The proposed overall network architecture.

The fusion attention block (FAB) is inserted into the U-Net backbone as a main building block. In the FAB, the input features first strengthen the attention to the information attenuation channels in channel attention block (CAB); after that, passed through Swin Transformer block (STB) and local attention block (LAB) to produce the non-local and local attention information, respectively; Finally, they are sent to the adaptive feature fusion block (AFFB) that performs the adaptive fusion of different feature information.

In the encoder part, input the given raw underwater image I∈RH×W×3, first perform 3×3 convolution to extract low-level features Fl∈RH×W×C, where H×W is the spatial dimension, and C is the number of channels. Then, the patch embedding layer maps it to the embedding features Fe∈R(H/2)×(W/2)×C. Following the design of U-net, the embedded features pass through K encoder stages. Each stage has a sequence of proposed FABs and a downsample layer. The encoder extracts image features at different scales and finally maps the input to deeper feature domains Fd∈R(H/16)×(W/16)×8C to strengthen semantic representation. The downsample layer reduces spatial dimensions and increases channels.

The bottleneck is located after the encoder, and it has a sequence of Fusion Attention blocks for increasing the aggregation of global information. This stage gets the depth feature output of the encoder Fd∈R(H/16)×(W/16)×8C, capturing global pixel dependencies and avoiding the result of excessive or insufficient enhancement.

The decoder also contains K stages, and each stage has a upsample layer and a sequence of proposed fusion attention blocks. Each stage receives information from both the previous stage and the skip-connection information of the corresponding encoder. It receives low-resolution depth features and gradually expands the resolution to reconstruct clear underwater images Fe∈R(H/2)×(W/2)×C. After that, the image reconstruction block reconstructs refined underwater images Fr∈RH×W×C and recovers spatial dimensions by upsampling. Finally, the network outputs an enhanced image I′∈RH×W×3 by using a 3×3 convolution layer to restore the scale of channels.

Fusion attention block

The inconsistently attenuation of underwater images on the space lead to the different degraded regions of the image should have different weights. Obviously, the convolution with the same treatment on the space does not accommodate this difference characteristic well.

The original Swin Transformer block is included in our proposed fusion attention block, where the self-attention mechanism allows the attention weights to interact with the image content, enabling flexible focus on any location in the input. STB limits the growth of computational complexity by using cyclic shift operations and masked MSA. The self-attention is computed within the non-overlapping local windows divided from the image. Two consecutive STBs are used as a whole unit. The first STB calculates self-attention within the window, and the second STB computes the self-attention within the shifted window. This strategy of shifting window enables the establishment of cross-window dependencies.

The calculation of the Swin Transformer block is expressed as

S^l=W−MSA(LN(Sl−1))+Sl−1Sl=MLP(LN(S^l))+S^l

where S^l and S are the output features of the W-MSA module and MLP. LN represents Layer Norm, while W-MSA and MLP refer to window-based multi-head self-attention and multilayer perceptron, respectively.

Our network is a hierarchical architecture, computing window self-attention on small size feature maps in deep networks, enabling modeling of long-range dependencies. The relative position embedding B∈R(2M−1)×(2M−1) is added to the attention calculation, where M represents the window size. The dot product attention calculation is expressed as

Attention(Q,K,V)= SoftMax(QKTC+B) ⋅V

where Q,K,V∈RC×HW represent the query, key, and value respectively. C refers to the channel dimension of the input feature, while SoftMax is multiple logistic regression.

Underwater images usually suffer from color shift since the water medium is biased to absorb certain wavelengths of light. Pixels in certain color channels of the images may have extremely low values, resulting in varying weights for different channel information. If we treat features of different channels equally, it cannot flexibly adapt to degraded underwater images.

We introduce a channel attention block (CAB) (Zhang et al., 2018) as shown in Fig. 2, which allows the model to capture weak degraded features. For the input feature map x∈RH×W×C, apply a global average pooling layer to generate a series of local descriptors z∈RH×W×C. Then, the first convolution compresses the scale of the channel, and after activation by the ReLU function, the second convolution restores the channels scale. The Sigmoid function activates multiple channel features to generate channel statistics s∈R1×1×C. The input is multiplied with this channel statistics to produce the feature map with weights in the channel dimension. The channel attention block is expressed as

Figure 2 Architecture of channel attention block.

z=GAP(x)r=ReLU(ConvD(z))s=Sigmoid(ConvU(r))x^c=sc⋅xc

where x represents the input feature map, GAP refers to the global average pooling layer, and ConvD and ConvU denote two convolution layers. ReLU and Sigmoid both represent activation functions. In the c-th channel, sc denotes the scaling factor, while xc represents the feature map.

Although Transformer can capture long-range dependencies well, it has insufficient ability to capture local features (Li et al., 2021; Wu et al., 2021) due to the lack of inductive bias of convolution. However, local context information is essential for image detail recovery. The local domain information of degraded pixels contributes to reconstructing clear images.

Therefore, we propose a local attention block (LAB), which is placed in another branch to capture useful local context information. The roles of the two branches are distinguished and exploit their respective advantages to enhance underwater images effectively. In Fig. 3, the local attention block comprises four residual blocks, and the calculation of every residual block as

Figure 3 Architecture of local attention block.

Xl=Conv(ReLU(Conv(Xl−1)))+Xl−1

where Conv denotes the convolution layer of 3×3. The input is activated by the ReLU layer after passing the first convolution, then enters the second convolution, and finally obtains a residual output. The input of the l-th residual block is Xl−1, and the output is Xl. We set l to 4.

We use Swin Transformer block (STB) to model remote dependencies and perform global attention while letting local attention modules capture local contextual information. To enable the adaptive collaboration between the global attention branch and local attention branch, we introduce an adaptive feature fusion block (AFFB) (Li et al., 2019b) that is learnable and allows dynamic adjusting weights of two branches.

The adaptive feature fusion block is illustrated in Fig. 4. First, incorporate the features of two branches by element addition, and then the global pooling layer is employed to obtain the global feature vector v. Following that, the dimensions are scaled down by the first linear layer, and then the two linear layers restore the scale and generate two weight vectors W1 and W2 to guide the selection of the two branches. Finally, the weights are multiplied with the input features separately, and the output results fuse the information on both branches. It allows the outputs of the FAB can be chosen flexibly between the two branches according to their features. The calculation of the AFFB is expressed as

Figure 4 Architecture of adaptive feature fusion block.

v=GAP(XSTL+XLA)v^=FCD(v)W1,W2=SoftMax([FCU1(v^),FCU2(v^)])F=XSTL⋅W1+XLA⋅W2

where XSTL and XLA denotes the output of STB and local attention block. FCD denotes the Linear layer with reduced dimensionality. FCU1 and FCU2 denotes two independent Linear layers. GAP stands for global average pooling layer, and SoftMax is a multiple logistic regression. W1 and W2 represent the weights on the two branches, respectively.

In summary, the process of FAB is defined as

X^=CA(X)Y1,Y2=STB(X^),LA(X^)Z=AFFB(Y1,Y2)

where X is the input of FAB, CA denotes channel attention block, STB and LA refer to Swin Transformer block and local attention block, and AFFB represents adaptive feature fusion block. Y1 and Y2 represent global attention and local attention, respectively, while Z represents the fused features outputted by the FAB. Our proposed FAB serves as the basic building block of the network, which has channel attention to strengthen the channel information with color degradation. To focus on degraded image regions and recover image details, it performs global attention and local attention in two parallel branches and adaptively fuses the attention results.

Loss function

To train our network, we employed a loss function that comprises two parts, Lcha and Lper. We use the robust Charbonnier loss (Lai et al., 2018) to learn the global similarity of image contents. Compared to L1 loss, it allows the function differentiable at zero by introducing a constant, which avoids the drawbacks of L1 loss. The Charbonnier loss calculation is shown below

Lcha=‖Iu−Xu‖2+ϵ2

where Iu represents the enhanced underwater image, Xu is the ground-truth image, and ε=10−3 refers to a constant value.

However, due to the neglect of the visual characteristics of the human eye by the Charbonnier loss, the generated results fail to maintain visual consistency, resulting in suboptimal visual effects. To enhance the subjective visual effects of images on human eyes, we introduce a perceptual loss in training. The perceptual structure of an image is represented by the pre-trained network VGG-19 (Simonyan & Zisserman, 2014). The perceptual loss Lper represents the distance between the two features of the enhanced image Iu and the ground truth image Xu output by the pre-training model, which is defined as

Lper=∑j=1N1CjHjWj‖ϕj(Iu)−ϕj(Xu)‖22

where N denotes the batch size. ϕj(⋅) indicates the output features of the jth layer VGG-19 network and CjHjWj denotes the dimensions feature map, namely the channel, height and width.

In summary, Charbonnier loss exhibits good robustness to outliers and possesses smooth penalties, making it suitable for image restoration tasks. While perceptual loss is based on extracted features, it can capture the semantic information of an image, making it suitable for learning the content of the image. The overall loss function integrating Lcha and Lper is expressed as

Lsum=λ1Lcha+λ2Lper

where λ1 and λ2 denote the importance coefficients, they are set to 1 and 0.1. They represent the balance between overall performance and visual effects.

Experiments

Underwater image synthesis

In contrast to some advanced vision tasks with large datasets available, underwater image enhancement tasks rely on synthetic datasets. The datasets used in the current related research can be classified into three categories. Firstly, using the unsupervised method of CycleGAN to generate images with underwater style. Images generated in this way need to select good results, and the dataset lacks the characteristics of various water types. Secondly, use publicly available datasets. However, these datasets are usually small, which can constrain the model's generalization ability if used for training. The last category is using indoor scene images with depth maps to synthesize simulated underwater images in accordance with the underwater image formation model.

We follow Zhou & Yan (2020) to synthesize the dataset for the training network. It describes three typical water types and provides the attenuation and background light coefficients. To simulate complex underwater scenes, the parameters provided have random values that increase the diversity of the dataset. The indoor dataset provided by Sun rgb-d (Song, Lichtenberg & Xiao, 2015) is adopted to synthesize underwater images in accordance with the underwater image degradation process defined in Eqs. (1) and (2).

Due to imperfections in some of the depth maps, we exclude flawed depth maps and select 7,680 indoor images with corresponding depth maps to synthesize a dataset. The dataset contains three types of underwater scenes consisting of 23,040 simulated underwater images and corresponding ground truth images.

Table 1 shows the normalized residual energy ratio Nrer and background light Bλ for three types of water. We set the scene depth d(x) from the nearest 0.5 m to the farthest 15 m (Anwar, Li & Porikli, 2018). To increase the computational efficiency of network training, we resized the synthesized underwater images to 256×256. Overall, the synthesized underwater image dataset is based on underwater image formation model and conforms to the principles of underwater imaging.

Table 1 Parameter settings of Nrer(λ) and Bλ for red, green and blue channels of three water types.

Type	Parameter	Red	Green	Blue	
B	Nrer(λ)	0.79 + 0.06*rand()	0.92 + 0.06*rand()	0.94 + 0.05*rand()	
Bλ	0.05 + 0.15*rand()	0.60 + 0.30*rand()	0.70 + 0.29*rand()	
C	Nrer(λ)	0.71 + 0.04*rand()	0.82 + 0.06*rand()	0.80 + 0.07*rand()	
Bλ	0.05 + 0.15*rand()	0.60 + 0.30*rand()	0.70 + 0.29*rand()	
D	Nrer(λ)	0.67	0.73	0.67	
Bλ	0.15	0.80	0.75	

Implement details

The proposed model is trained using the synthetic dataset and set the image size of the input network to 256×256. The network training utilizes 22,610 images from the synthetic dataset, while the validation set contains 430 images. Our network is implemented on Ubuntu20 with Intel(R) Xeon E7-4809 v3 CPU, NVIDIA RTX 3090 GPU, and PyTorch framework. Training with Adam (Kingma & Ba, 2014) optimizer, which β1 and β2 set to 0.9 and 0.999. We configured the batch size at 2 and set the learning rate at 0.0005 and a decay rate of 0.5 every 50 epochs, training 200 epochs.

Synthetic underwater images evaluation

First, our proposed method is evaluated on synthetically underwater images. To demonstrate its superior performance, we compared proposed network with seven competition methods. It includes three traditional methods (UDCP (Drews et al., 2013), Fusion (Ancuti et al., 2012), RGHS (Huang et al., 2018)), three CNN-based methods (UWCNN (Anwar, Li & Porikli, 2018), FUnIE (Islam, Xia & Sattar, 2020), WaterNet (Li et al., 2019a)) and one Transformer-based method (U-shape (Peng, Zhu & Bian, 2023)). The results of the comparison method are obtained from executing the code released by the corresponding author.

Our synthetic datasets were subjected to comprehensive experiments. In the dataset, test images are denoted as Test-S and training images are denoted as Train-U. The PSNR and SSIM quality measurement are employed for full-reference evaluation on synthetic datasets with reference images. The full-reference evaluation metric indicates its proximity to the reference image. PSNR evaluates the image quality from the perspective of the loss of signal. A higher PSNR value indicates a higher quality reconstructed image with more similar image content. The SSIM measurement takes into account luminance, contrast and structure, which is comparable to the human visual system. It has a value in the range of 0 to 1, with higher values meaning a more similar image structure.

The quantitative evaluations as shown in Table 2, while qualitative comparisons results on Test-S are demonstrated in Fig 5. Our model has the best results on PSNR and SSIM metrics, as indicated in Table 2. Its 20.46 PSNR and 0.82 SSIM on Test-S are higher than the suboptimal method by 0.1 PSNR and 0.01 SSIM. Moreover, the proposed method has a moderate number of parameters, and low computational cost during training.

Table 2 The quantitative evaluations of different methods on synthetic dataset.

Training	Methods	Test-S	Efficiency	
PSNR↑	SSIM↑	Parameters(M)↓	FLOPs(G)↓	
Train-U	UDCP	11.57	0.57	–	–	
Fusion	19.61	0.79	–	–	
RGHS	18.61	0.80	–	–	
UWCNN-typeI	17.51	0.75	0.04	0.79	
Water-Net	19.25	0.79	1.09	142.91	
FUnIE	15.48	0.72	7.69	10.68	
U-shape	20.36	0.81	65.60	66.20	
Ours	20.46	0.82	11.30	13.56	
Note:

The optimal result is represented in bold.

Figure 5 (A–I) Qualitative comparisons of different methods on synthetic dataset Test-S.

(A) Raw images. (B) Fusion (Ancuti et al., 2012). (C) RGHS (Huang et al., 2018). (D) UWCNN-I (Anwar, Li & Porikli, 2018). (E) Water-Net (Li et al., 2019a). (F) FUnIE (Islam, Xia & Sattar, 2020). (G) U-shape (Peng, Zhu & Bian, 2023). (H) Our method. (I) Ground truth. The visual results show that the results of our method recover the color of image without color artifacts and has the best image detail texture. Image source credit: Hou et al. (2020) CC BY 4.0.

The enhanced outcomes of synthetic underwater images of the test set, which includes images of degraded underwater scenes in dark green, are displayed in Fig. 5. Compared to competing methods, the images enhanced by our method exhibit the highest similarity to the ground truth. The traditional methods have limited ability to correct the color, which is only for images with color shifts slightly. Fusion overcorrects image colors resulting in unnatural tones, while RGHS cannot correct degraded image colors. The deep learning approach enhances images by extracting image features, which obtain better results than traditional methods. Since UWCNN-I uses one type of water dataset for training, it leads to a model with insufficient generalization ability and produces poor image enhancement results. Water-Net and FUnIE remove most color shift effects. However, the resultant image of Water-Net has poor details and low color saturation, while FUnIE recovers the colors incompletely to produce color artifacts and low image contrast. U-shape removes the color shift effect but causes over-enhancement of colors and a lack of ability to reconstruct image detail. The proposed method has sharper enhancement results with better color and detail reconstruction.

Real underwater images evaluation

Furthermore, our model was evaluated on real-world underwater images to validate its generalization. Likewise, our model is compared with the seven competing approaches to observe enhancement effects on real-world underwater images. These images utilized for the evaluation are taken from the available public datasets.

We perform evaluations on real underwater images Test-C60 (Li et al., 2019a) and Test-M to verify actual effectiveness of our model. Test-C60 Contains 60 challenge images, which are difficult to enhance by existing methods. Test-M contains 515 underwater images including underwater marine scenes and marine animals. Since some of the images did not have corresponding reference images, we have employed non-reference quality evaluation measure such as UIQM (Panetta, Gao & Agaian, 2015), UCIQE (Yang & Sowmya, 2015), NIQE (Mittal, Soundararajan & Bovik, 2012b) and BRISQUE (Mittal, Moorthy & Bovik, 2012a). Among them, higher UIQM and UCIQE scores indicate that the result has better visual perception. UIQM evaluates image quality from aspects of color, sharpness and contrast. Also, UCIQE measurement from aspects of image hue, saturation and contrast. A lower NIQE score indicates higher visual quality, which measures the difference in image features over a multivariate distribution constructed from features of natural images. BRISQUE evaluates the quality of an image based on the natural scene statistics present in the image, including the gradient magnitude, contrast, and brightness.

The quantitative evaluations as shown in Table 3, while qualitative comparisons are demonstrated in Figs. 6 and 7. According to the statistical results, the FUnIE achieved the highest UIQM and NIQE scores on Test-C60. However, the results of FUnIE display drawbacks with low color saturation, color artifacts and insufficient detail recovery. Due to the inability of certain evaluation metrics to differentiate perceptual differences, even if better evaluation values are achieved, the subjective visual effects of the results may still exhibit color distortions. This could be attributed to the sensitivity of the evaluation metrics to the types of distortions present. For instance, UIQM and UCIQE are insensitive to color artifacts and have biases for certain features (Berman et al., 2020). Moreover, in the no-reference evaluation of underwater images, non-data-driven methods generally outperform data-driven methods. From this perspective, the proposed method performs relatively well among deep learning-based methods, with the exception of the FUnIE results. In summary, the quantitative evaluation cannot determine the final performance, and a visual comparison is necessary.

Table 3 The quantitative evaluations of different methods on synthetic dataset.

Methods	Test-C60	Test-M	
UIQM↑	UCIQE↑	NIQE↓	BRISQUE↓	UIQM↑	UCIQE↑	NIQE↓	BRISQUE↓	
UDCP	1.32	0.55	6.76	1,355.49	1.82	0.60	5.65	16,114.95	
Fusion	2.56	0.62	5.77	855.27	2.76	0.62	5.19	15,731.40	
RGHS	1.96	0.61	5.79	1,068.12	2.30	0.61	5.31	15,822.46	
UWCNN-typeI	2.39	0.50	6.14	1,609.54	2.92	0.55	5.40	17,996.88	
Water-Net	2.09	0.54	7.54	2,042.68	2.28	0.58	5.67	18,138.59	
FUnIE	2.94	0.52	5.43	821.39	3.18	0.56	5.15	17,618.61	
U-shape	2.49	0.54	6.78	1,771.79	2.91	0.58	5.61	18,547.88	
Ours	2.45	0.55	7.19	1,591.15	2.62	0.60	5.37	17,304.16	
Note:

The optimal and suboptimal results are represented in bold and underline styling.

Figure 6 Qualitative comparisons of different methods on the Test-C60.

(A) Raw images. (B) UDCP (Drews et al., 2013). (C) Fusion (Ancuti et al., 2012). (D) RGHS (Huang et al., 2018). (E) UWCNN-I (Anwar, Li & Porikli, 2018). (F) FUnIE (Islam, Xia & Sattar, 2020). (G) Water-Net (Li et al., 2019a). (H) U-shape (Peng, Zhu & Bian, 2023). (I) Our method. Image source credit: © UIEB Dataset, https://li-chongyi.github.io/proj_benchmark.html, all rights reserved.

Figure 7 Qualitative comparisons of different methods on the Test-M.

(A) Raw images. (B) UDCP (Drews et al., 2013). (C) Fusion (Ancuti et al., 2012). (D) RGHS (Huang et al., 2018). (E) UWCNN-I (Anwar, Li & Porikli, 2018). (F) FUnIE (Islam, Xia & Sattar, 2020). (G) Water-Net (Li et al., 2019a). (H) U-shape (Peng, Zhu & Bian, 2023). (I) Our method. Images source credit: Xu Liu, December 15, 2022, “MIX Benchmark”, IEEE Dataport, doi: https://dx.doi.org/10.21227/bmkp-f773.

Figure 6 shows the visual effects of the comparison approaches on Test-C60. It is evident that the traditional approaches produce unsatisfactory enhancement results. UDCP introduces unnatural colors and the image has low brightness. While the results of Fusion and RGHS have color artifacts. The data-driven methods are better than the traditional methods in general, but there are also some defects. The results of UWCNN-I have the problem of color casts. FUnIE and Water-Net result in images with low color contrast. The method based on Transformer obtains enhanced images with better color restoration. However, U-shape produces relatively blurry results, while the proposed method has better image detail reconstruction ability and image color contrast.

Figure 7 shows the visual effects of the different methods on Test-M. As can be seen, the traditional methods cannot effectively deal with the color shift problem. For background areas with more backscattering, Fusion and RGHS cannot correct the color. And UDCP over-enhances and introduces unnatural colors. FUnIE corrects the color to some extent, but color artifacts remain. The dehazing effects of UWCNN-I are not satisfactory, while the enhanced results of Water-Net lack color saturation. The Transformer-based method has better visual results since it solves the color shift problem more effectively than other methods. Compared to the U-shape, the results of our method have better color saturation and sharper details.

In our network, the fusion attention block can strengthen the attention to degraded spatial regions and color channels, thus obtaining results without color bias and artifacts. It also extracts image features by utilizing global and local attention to reconstruct the images with high-quality details.

Ablation study

In order to prove the effectiveness of every block in our model, we conducted an ablation study. The factors we consider include networks without channel attention block (CAB), networks without local attention block (LAB) and adaptive feature fusion block (AFFB) and networks without perceptual loss Lper.

All experiments were trained by Train-U. Table 4 reports the average statistics on Test-S. Where full model suggested in this article achieves the best results. Therefore, this reflects the effectiveness of the fusion attention block of our model. Table 5 reports the influence of different training losses. As shown in Table 5, with the addition of the loss function, the performance of the proposed network has improved, indicating the effectiveness of introducing perceptual loss.

Table 4 Statistical results of the proposed method with different variants.

Methods	CAB	AFFB	LAB	Test-S	
PSNR↑	SSIM↑	
Only CAB	✓	✗	✗	19.26	0.79	
AFFB+LAB	✗	✓	✓	20.20	0.80	
Full model	✓	✓	✓	20.46	0.82	
Note:

The optimal result is represented in bold.

Table 5 Statistical results of the different training losses.

Loss functions	Lcha	Lper	Test-S	
PSNR↑	SSIM↑	
w/o Lper	✓	✗	20.01	0.80	
Lsum	✓	✓	20.46	0.82	
Note:

The optimal result is represented in bold.

As shown in Fig. 8, the result of only CAB has more noise due to the lack of local detail reconstruction ability of LAB. While AFFB + LAB avoids more noise generation but produces results with hue shifting. Overall, our full model produces the highest-quality images, which are closest to the ground truth image. These results demonstrate that the FAB and loss function used in the proposed method significantly contribute to achieving the final visually pleasing results.

Figure 8 Visual results of the ablation study.

(A) Raw synthetic underwater images. (B) Only CAB. (C) AFFB+LAB. (D) w/o Lper. (E) Full model. (F) Ground truth. Image source credit: Hou et al. (2020).

Conclusion

We presented a dual attention fusion Transformer for underwater image enhancement. It combines the non-local modeling ability of the Transformer with the local modeling ability of convolution and can adaptively fuse their features. The network introduces the Swin Transformer block (STB) to solve the spatial attenuation inconsistency problem of underwater images, while its local attention block (LAB) is designed to recover the texture information of images. The function of adaptive feature fusion block (AFFB) is to perform adaptive fusion of non-local and local feature information, which is beneficial for network performance. Since underwater images are severely attenuated in some color channels, the channel attention block (CAB) aimed to focus on important channel feature information. In addition, we synthesize an underwater image dataset in accordance with the underwater image formation model, which contains scene images of multiple water types. Experiments demonstrate that this dataset is effective for training our model. The training incorporates a perceptual loss function to improve the visual perception of the reconstructed images. Through comprehensive experiments on synthetic and real underwater images, our network is validated to have excellent capabilities in correcting image colors and recovering image details. Overall, compared to other methods, the proposed approach generates relatively good results based on a comprehensive analysis of qualitative and quantitative experiments. However, due to the lack of prior information in the underwater image enhancement process, it becomes challenging for enhancement algorithms to distinguish between real details and noise in the image. In certain specific environments, the proposed network cannot achieve satisfactory results. In the future, we will attempt to incorporate the physical formation into the model to design a more efficient underwater image enhancement network model.

Supplemental Information

Supplemental Information 1 Network model and training code.

Supplemental Information 2 Raw data.

Experimental results of images on the test set.

Additional Information and Declarations

Competing Interests

Author Contributions

Data Availability

The authors declare that they have no competing interests.

Xianjie Hu conceived and designed the experiments, performed the experiments, performed the computation work, authored or reviewed drafts of the article, and approved the final draft.

Jing Liu conceived and designed the experiments, analyzed the data, prepared figures and/or tables, and approved the final draft.

Heng Li analyzed the data, performed the computation work, authored or reviewed drafts of the article, and approved the final draft.

Hui Liu performed the experiments, performed the computation work, prepared figures and/or tables, and approved the final draft.

Xiaojun Xue conceived and designed the experiments, analyzed the data, authored or reviewed drafts of the article, and approved the final draft.

The following information was supplied regarding data availability:

The network model is available in the Supplemental File and the dataset for training the network is available at GitHub and Zenodo:

- https://github.com/XianjieH/UDAFT.

- xianjie H. (2023). XianjieH/UDAFT: First release (v1.0.0). Zenodo. https://doi.org/10.5281/zenodo.10347349.

The MIX BENCHMARK Dataset is available at: https://ieee-dataport.org/documents/mix-benchmark.

The An Underwater Image Enhancement Benchmark (UIEB) Dataset and Beyond is available at: https://li-chongyi.github.io/proj_benchmark.html.

The SUN RGB-D: A RGB-D Scene Understanding Benchmark Suite Dataset

is available at: https://rgbd.cs.princeton.edu/.

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
