# Peer review of "An effective transformer based on dual attention fusion for underwater image enhancement"

_PeerJ Computer Science, doi:10.7717/peerj-cs.1783_

## Round 0.1 · original submission · Major Revisions

Thanks for your interest in the journal. I look forward to your revised version with an explanation of how you addressed the first reviewer's comments.

Reviewer 1 ·

Basic reporting

This paper studies the problem of underwater image enhancement. The main idea is using deep learning based on dual attention fusion model. This idea is not sufficiently novel, because the authors only adopted this model and introduced perceptual loss function for improved visual perception. The strong aspect of the paper is the authors show that the proposed technique to achieve better quality processing than the enhancement methods.
I have the following problems with this paper:
- The innovation of this paper is not clear, and it is difficult for readers to understand the main contributions of this paper. The Introduction can be revised to emphasize the main contribution of the work.
- The article has a poor organization: the quality of illustrations can be improved; it is necessary to include a block diagram explaining the proposed algorithm.
- In using formulas for the detailed explanation, this article does not explain all the letters involved; please pay attention to the letter interpretation.
- The proposed fitness function needs to be described in more experiment detail too.

Experimental design

- The authors did not a present full objective quality comparison on test images to check the quality of their enhancement methodology. The methodology of using full reference measures (SSIM, PSNR) more suitable for image denoising problem. I recommend presenting an estimation of non-reference measures whit more specified for image enhancement (for example, AME, EMEE, SDME, Visibility, TDME, BIQI, BRISQUE, ILNIQE).
- Several technical information and details about computation costs are required.
- For the experiments, some discussion should be given about the results to explain the merits and drawbacks of the proposed method.

Validity of the findings

no comment

Reviewer 2 ·

Basic reporting

This article is written in professional English. Expressions are clear and unambiguous. Ample background works are provided.

Experimental design

As the research question is well-define, they introduce the proposed architecture clearly. They also did a very extensive experiment and obtain very promising results in multiple quality evolution measures.

Validity of the findings

Authors presented a dual attention fusion Transformer for underwater image enhancement. According to the results of experiments, it is validated that the proposed architecture improves the quality of underwater images and outperforms most of the compared algorithms. They also proposed a synthesized underwater image dataset, which contains images of various water types.

---

## Round 0.2 · accepted · Accept

The paper has been revised sufficiently and is ready to go for publication.

Reviewer 1 ·

Basic reporting

The authors have addressed all the questions I have mentioned. I have no further questions.

Experimental design

The strong aspect of the paper is the authors show that the proposed technique to achieve better quality processing than the enhancement methods.

Validity of the findings

All underlying data have been provided.

Additional comments

n/a